# Assessment of Intra-Urban Heat Island in a Densely Populated City Using Remote Sensing: A Case Study for Manila City

Mark Angelo Purio [1,2,*], Tetsunobu Yoshitake [3] and Mengu Cho [1]

1. Laboratory of Spacecraft Environmental Interaction Engineering, Kyushu Institute of Technology, Kitakyushu 804-8550, Japan
2. Electronics Engineering Department, Adamson University, Manila 1000, Philippines
3. Department of Civil and Architectural Engineering, Kyushu Institute of Technology, Kitakyushu 804-8550, Japan
* Correspondence: purio.mark-angelo894@mail.kyutech.jp or markangelo.purio@adamson.edu.ph; Tel.: +63-8-524-2011

**Abstract:** Changes in the environment occur in cities due to increased urbanization and population growth. Sustainable Development Goal (SDG) 11 is intrinsically linked to the environment, one facet of which is the need for universal access to secure, inclusive, and accessible green and public places. As urban heat islands (UHI) have the potential to negatively influence cities and their residents, existing resources and data must be used to identify and quantify these effects. To address this, we present the use of satellite-derived (2013–2022) and meteorological data (2014–2020) to assess intra-urban heat islands in Manila City, Philippines. The assessment includes (a) understanding the temporal variability of air temperature measurements and outdoor thermal comfort based on meteorological data, (b) comparative and correlative analysis between common Land-Use Land Cover indicators (Normalized Difference Vegetation Index (NDVI), Normalized Difference Water Index (NDWI) and Normalized Difference Built-up Index (NDBI)) and Land Surface Temperature (LST), (c) spatial and temporal analysis of LST using spatial statistics techniques, and (d) generation of an intra-urban heat island (IUHI) map with a recommended class of action using a suitability analysis model. Finally, the areas that need intervention are compared to the affected population, and suggestions to enhance the thermal characteristics of the city and mitigate the effects of UHI are established.

**Keywords:** intra-urban heat island; remote sensing; space-time; GIS; SDGs

## 1. Introduction

More focus has been placed on global urbanization recently as more people around the globe move to urban areas every year. Today, more than half of the world's population resides in urban areas, and forecasts indicate that an increasing share of urban residents will be responsible for almost all future population increases. The complicated socioeconomic process of urbanization affects the built environment, relocating the population's spatial distribution from rural to urban regions, and converting once rural areas into urban ones. It has an impact on dominant occupations, lifestyles, cultures, and behaviors in both urban and rural regions, altering both the demographic and social structure. The key effects of urbanization include the quantity, size, and density of urban settlements as well as the population share between urban and rural inhabitants [1,2]. By ensuring that cities and human settlements are inclusive, safe, and resilient, SDG 11—one of the United Nations' "17 Sustainable Development Goals"—highlights the importance that cities play in the world's political agenda [3]. In the review of Estoque [4], despite initial efforts, the UN Global Sustainable Development Report 2019 [5] found that the world is not on track to achieve most SDG objectives including indicators related to SDG 11.

Due to urban regions developing more quickly with population expansion, environmental changes ensue [6]. Loss of open space and animal habitats, water and air pollution, transportation, health concerns, and agricultural capacity are a few implications, while changing thermal properties is another result of urbanization and city growth. In terms of the increase in the Land Surface Temperature (LST) of the landscape, ongoing urbanization and the growth of impermeable surfaces are both factors [7,8]. As urban regions expand, the topography changes. Buildings, roads, and other forms of infrastructure take the place of open space and plants, for example, and permeable and moist surfaces eventually become impermeable and dry [9].

As a result of this development, urban heat islands (UHI) occur—a phenomenon in which urban areas experience warmer temperatures than their rural surroundings [10]. In particular, densely packed structures with little greenery develop "islands" with greater temperatures than their surroundings [9,11–13]. UHI may influence the increased risk of health-related conditions, increase in energy consumption, elevated pollutants, and water quality [14]. Urban heat islands (UHI) have the potential to have a detrimental impact on cities and their inhabitants, and as such, available resources and data must be used to detect and quantify these consequences. SDG 11 works toward making societies more sustainable and resilient by giving us a unique chance to make sure that the infrastructure we build today will still be useful in the future. This can be done by investing in parks and green spaces in cities, which will help reduce the "urban heat island effect" [3].

Aside from this, according to a growing body of research [15–17], "intra-urban" heat islands (IUHI), or regions within a city that are hotter than others due to an unequal distribution of heat-absorbing buildings and pavements, as well as cooler zones with trees and greenery, are becoming more prevalent [18]. Intra-Urban Heat Islands (IUHI) detection is of major interest to city planners since high temperatures influence energy usage and human health [16]. In 2015, Martin et al. [19] referred to surface intra-UHI as the detection of hotspots in a metropolis which is made possible by determining temperature thresholds by spatial reference. Consequently, the data can then be used to identify regions of interest in a city and potentially trigger alarms at a finer spatial scale. An example is a study conducted by Igergård et al. [20] in the Stockholm municipality.

In the literature, remote sensing is a good resource to understand the link between urban expansion and the characteristics describing the thermal changes in both geographical and temporal contexts [7,14,21]. Among remote sensing data, satellites are used more to estimate LST due to the thermal and passive microwave sensors aboard them. Although satellite data are very useful, Zhou et al. [14] stressed in their systematic review that retrieved satellite LST and air temperature differences, the effect of clouds, spatial and temporal resolution trade-off, SUHI quantification methods, varying land use land cover methods, and SUHI accuracy assessment are among the current challenges faced by UHI researchers. Worse, the limited availability of datasets for SUHI studies and applications exacerbates the challenges.

The increasing number of publications on the effect of UHI, particularly after 2016, reflects the scientific community's interest in disseminating information about this subject, which investigates its causes and ramifications from several viewpoints, including environmental, social, and economic [22]. The Philippines, like the rest of the world, is experiencing fast urbanization and a population density increase. Furthermore, these densely populated cities are largely clustered in Metro Manila [23,24]. In this context, statistically analyzing satellite data geographically and temporally, Landicho and Blanco [25] confirmed that intra-urban heat islands (IUHI) in Metro Manila are prevalent in 2019 while Alcantara et al. [26,27] conducted UHI studies in Quezon City. Estoque et al. [28], moreover, used satellite-derived surface temperature data and socio-ecological factors to analyze the present health risk in 139 Philippines cities. In addition, cities outside of Metro Manila were part of the Project GUHeat [24], which conducted urban heat island studies in cities such as Baguio [29], Cebu [30], Davao [31], Iloilo [32], Mandaue [33], and Zamboanga [34].

Given prior geographic biases in the literature, greater attention should be placed on understudied areas or cities, as proposed by Zhou et al. [14] and Almeida et al. [22] in their reviews. Furthermore, little published research explores how UHI affects the population because of a lack of fine-scale geographic population data [35]. Consequently, as there is inadequate research about UHI conducted in the country, area-specific assessment in cities like Manila would provide further details on how changes in the landscape impact the city's heat situation and will serve as a basis for urban planners and policymakers for mitigation and improvement. This also supports the goals of SDG 11 to aid the futureproofing of infrastructures for cleaner and greener cities.

The novelty of the present work is the use of space-time pattern mining to assess the presence of intra-urban heat islands using remote sensing data. Although this type of methodology is well established for space-time analysis applications, its usage on remote sensing data such as land surface temperature has not been extensively studied. Moreover, according to the author's knowledge, no work was dedicated to including the population and settlement data in such an assessment method for Manila City or any highly urbanized cities in the Philippines.

Its main purpose is to use satellite-derived and in situ meteorological remote sensing data to assess the presence of intra-urban heat islands in Manila City. Moreover, demographic data such as population and settlement data were used to enhance the assessment. Data represented in a space-time cube were used to carry out a space-time pattern mining approach in generating an Intra-Urban Heat Island (IUHI) map for Manila City. Finally, city-specific strategies to promote outdoor thermal comfort and hotspot interventions were also suggested. This paper is divided into five sections:

- Section 1 introduces the research, the state-of-the-art review, research gaps, and a statement of purpose.
- Section 2 presents the data and a detailed discussion of the methods employed.
- Section 3 shows the description of the results and output of the analysis.
- Section 4 discusses the results in detail, interprets the findings concerning previous studies, and examines the context of the outcomes of the study.
- Section 5 summarizes what was done in the study, the findings, and future work.

## 2. Materials and Methods

### 2.1. Study Area

As shown in Figure 1, Manila City is located in the northern Philippines archipelago, on the island of Luzon, on the eastern side of the old Manila Bay, with the Pasig River running through it [36,37]. As the Philippines' capital, Manila is considered to have the highest population density among the country's highly urbanized cities, and even among the world's densest cities. In 2020, the Philippine Statistics Authority [38] recorded that 1.84 million population reside in its 24.98 square kilometer land area, which translates to about 74,000 inhabitants per square km. The spatial attributes of the city are shown in Table 1.

**Table 1.** Spatial Attributes of Manila City.

| Data Attributes | Description |
| --- | --- |
| Spatial Reference | GCS WGS 1984 |
| Spatial Resolution | Approximately 30 m |
| Number of Pixels Covered | 48,667 |
| Data Format | Geo tiff |

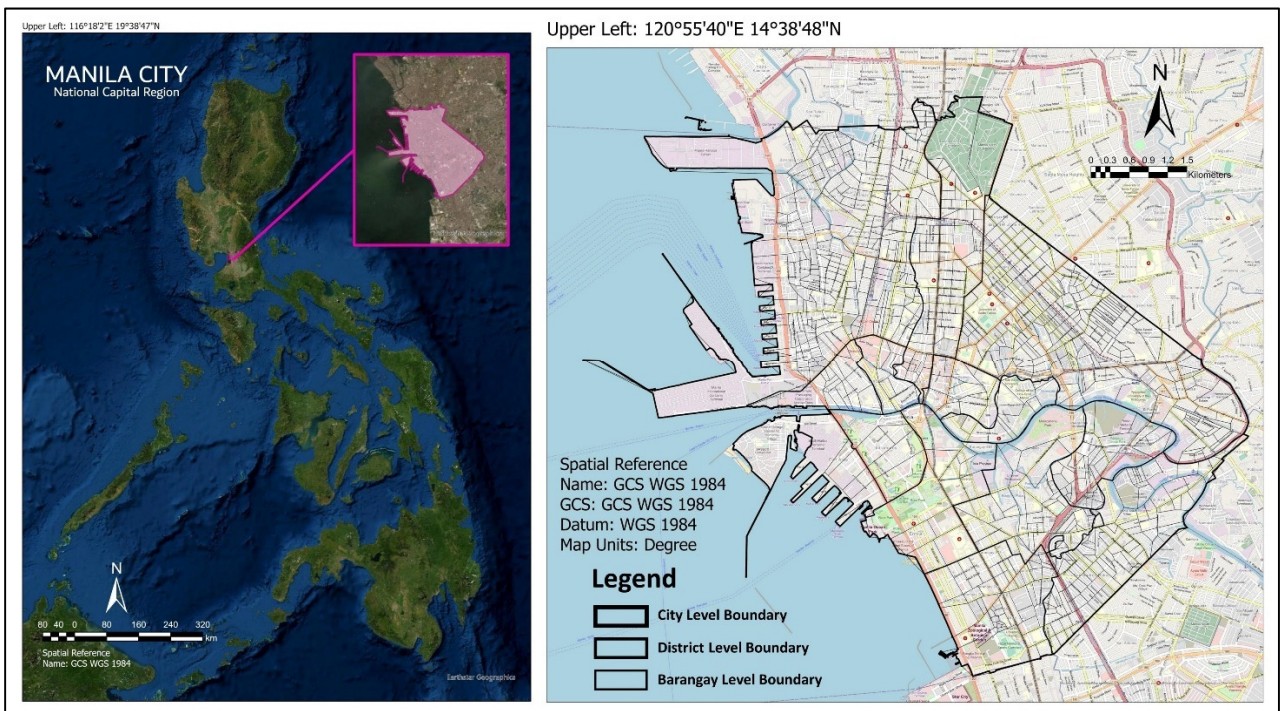

**Figure 1.** Manila City's geographical location (**left**) and administrative boundary (**right**).

According to the Koppen Climate Classification [39] Manila has a tropical rainforest climate (Af). There is no dry season in a tropical rainforest environment, and it rains at least 60 mm per month throughout the year (2.36 in). Tropical rainforest climates do not have distinct seasons; it is hot and humid year-round, with frequent and heavy rains. Manila has an annual average temperature of 27.8 degrees Celsius, or 82.0 degrees Fahrenheit. With an average temperature of 85.0 °F (29.4 °C), April is the hottest month of the year, while the lowest month is January at 79.0 °F (26.1 °C) [39].

### 2.2. Data and Data Sources

This section enumerates the data and their sources including their descriptions, attributes, and the methods employed to obtain and prepare the data.

#### 2.2.1. Manila City Administrative Boundary

An administrative boundary represents subdivisions of areas, territories, or jurisdictions recognized by governments for administrative purposes [40]. The Philippines follows the Philippine Standard Geographic Code (PSGC) with different geographic levels such as region, province, city/municipality, and the smallest unit, barangay [41]. For the research, we need the shapefiles for Manila City at the city, district, and barangay levels. A published GitHub repository [42] was used since it is complete with all the needed geographic levels projected using the WGS 1984, latitude/longitude projection. These shapefiles were sourced from reliable webpages such as the OCHA Services Website [43] and GADM.org [44].

#### 2.2.2. In-Situ Meteorological Data

Meteorological raw data taken daily from 2014 to 2018 were provided by the weather bureau of the Philippines. The meteorological parameters include rainfall amount, mean temperature, maximum temperature, minimum temperature, wind speed, wind direction, and relative humidity. Since just one synoptic station is in Port Area, Manila (14.5878°N latitude and 120.9690°E longitude), only point data are available for Manila City.

### 2.2.3. Population Data

Population density is a key metric for assessing domestic living circumstances. Due to the statistical approach used, traditional census statistics cannot represent the population's geographical distribution with a high degree of precision [35]. A high-resolution map estimate of the population density inside 30-m grid tiles was supplied by Data for Good Meta, which we used in this research. In this study, the population density demographic data for the year 2018 was used to give an insight into the distribution of people affected by the intra-urban heat island in Manila City. Since the downloaded data represent the whole country, we used ArcGIS Pro software to clip the region of interest based on the administrative boundary of Manila City. Aside from population density, those pixel grids with data are considered settlement areas while empty grids denote non-settlements areas in the city. Each cell's value represents the population density of that pixel/grid. This density may be expressed as a grid's area.

### 2.2.4. Satellite Data

Satellite-derived remote sensing data in the study were taken from MODIS and Landsat 8 satellite data products. Daily land surface temperatures (day and night) were obtained from MODIS between 2014 to 2018 as complementary data for the meteorological data mentioned above. Consequently, spatial yearly data raster for land surface temperature and spectral indices were downloaded from Landsat 8.

- MODIS Land Surface Temperature Product

Land Surface Temperature data were derived from the Collection-6 MODIS Land Surface Temperature product to complement the available meteorological data at hand. Details of their retrieval were reported in [45]. In this study, data were retrieved to complement the meteorological data since global daily LST data can be obtained with this. Although the spatial resolution is low, the temporal resolution of the MODIS dataset is good and was deemed applicable for the correlation analysis presented in Ref [45].

- Landsat 8 Data Product

The Climate Engine web app (https://app.climateengine.com/climateEngine# (accessed on 9 February 2020)) [46] was used to download analysis-ready Landsat 8 data which were preprocessed using the Google Earth Engine [47] platform. The web application allows easy download of Landsat Bands, Spectral Indices, and Land Surface Temperature aggregated per year of study. In ecological studies, digital numbers and reflectance are the most used while studies involving thermal bands often use digital numbers and temperatures. For this study, we used top-of-atmosphere (TOA) reflectance products to obtain the land surface temperature (LST) and surface reflectance (SR) products for spectral indices such as Normalized Difference Vegetation Index (NDVI), Normalized Difference Water Index (NDWI), and Normalized Difference Built-up Index (NDBI).

- Land Surface Temperature

According to ESA [48], "Land Surface Temperature (LST) is the radiative skin temperature of the land derived from solar radiation. A simplified definition would be how hot the "surface" of the Earth would feel to the touch in a particular location. From a satellite's point of view, the "surface" is whatever it sees when it looks through the atmosphere to the ground. It could be snow and ice, the grass on a lawn, the roof of a building, or the leaves in the canopy of a forest. Land surface temperature is not the same as the air temperature that is included in the daily weather report."

Landsat 8 passes the equator at 10:00 am $+/-$ 15 min (mean local time) [49] so the maps that will be generated are only based on measurements from this specific time of the day. While IUHI can be measured better in Manila City in the afternoon than in the morning, the limitation of satellite data to provide this led the researchers to use such Landsat data for the investigation. Deilami et al. [50] stressed in their review that the popularity of Landsat images for UHI studies can be attributed to factors such as being freely available

to researchers, their worldwide coverage with a reasonable spatial resolution of 30 × 30 m, and the long-term temporal coverage which enables researchers to extract the required information over a long period to monitor changes. Moreover, in their review article, about 22% of the papers reviewed use Landsat 8 data for UHI investigation with data available from 2013 to the present [50].

Raster data of land surface temperature data were taken from 2013 to 2022 on a yearly interval. Because of constraints in maximum cloud cover, LST within the year was obtained to depict maximum temperatures occurrence for that year. The top-of-atmosphere (TOA) product was used to illustrate the presence of cold and hotspots in the yearly intra-urban heat island map generated. Although the actual resolution Landsat 8 LST is 100 m, the analysis product downloaded from the climate engine is provided at 30 m.

- Normalized Difference Vegetation Index (NDVI)

The NDVI is a dimensionless index that describes the difference between visible and near-infrared reflectance of vegetation cover and can be used to estimate the density of green on an area of land. No green leaves produce a value near zero, yet calculations of NDVI for a particular pixel always yield a figure that falls between a negative one (−1) and a positive one (+1). A value of zero denotes no vegetation, whereas a value of close to one (0.8–0.9) represents the greatest potential density of green leaves [51]. The following formula is used to calculate NDVI:

$$\text{NDVI} = \frac{(\text{NIR} - \text{Red})}{(\text{NIR} + \text{Red})} \tag{1}$$

For Landsat data, NDVI $=$ (Band 5 $-$ Band 4/(Band 5 $+$ Band 4). This can be directly downloaded from the climate engine. Table 2 shows the ranges of NDVI and their corresponding land use land cover (LULC) classification.

**Table 2.** NDVI ranges for LULC Classification.

| NDVI Ranges | Land Use Land Cover (LULC) Classification | Class |
| --- | --- | --- |
| −1.0 to 0.0 | Water Body | 1 |
| 0.0 to +0.2 | Urban Built-up | 2 |
| +0.2 to +1.0 | Vegetation | 3 |

- Normalized Difference Water Index (NDWI)

NDWI is a measure of liquid water molecules in vegetation canopies that interacted with the incoming solar radiation. It is less sensitive to atmospheric scattering effects than NDVI [52]. This index uses NIR and SWIR bands where the resulting value ranges from minus one (−1) to plus one (+1). Positive values of NDWI correspond to high vegetation water content and high vegetation fraction cover. Negative NDWI values correspond to low vegetation water content and low vegetation fraction cover. In a period of water stress, NDWI will decrease. The following formula gives the NDWI value.

$$\text{NDWI} = \frac{(\text{NIR} - \text{SWIR1})}{(\text{NIR} + \text{SWIR1})} \tag{2}$$

For Landsat data, NDWI $=$ (Band 5 $-$ Band 6)(Band 5 $+$ Band 6). This can be directly downloaded from the climate engine. Table 3 shows the ranges of NDWI values and the corresponding water content classification.

**Table 3.** NDWI ranges for Water Content Classification.

| NDWI Ranges | Water Content Classification | Class |
| --- | --- | --- |
| −1.0 to 0.0 | Low Water Content | 1 |
| 0.0 to +0.1 | High Water Content | 2 |

- Normalized Difference Built-up Index (NDBI)

The Normalized Difference Built-up Index (NDBI) uses the NIR and SWIR bands to emphasize constructed built-up areas. It is a ratio based on mitigating the effects of terrain illumination differences as well as atmospheric effects [53,54]. A negative value of NDBI represents water bodies whereas a higher value represents build-up areas. NDBI value for vegetation is low. The following formula gives the NDBI value.

$$\text{NDBI} = \frac{(\text{SWIR1} - \text{NIR})}{(\text{SWIR1} + \text{NIR})} \tag{3}$$

For Landsat 8 data, NDBI $=$ (Band 6 $-$ Band 5)/(Band 6 $+$ Band 5). This cannot be directly downloaded from the climate engine, so the individual NIR and SWIR1 bands were downloaded, then NDBI was calculated using the raster calculator tool in ArcGIS Pro. Table 4 shows the ranges of NDBI values and the corresponding build-up area classification.

**Table 4.** NDBI ranges for Build-up Area Classification.

| NDBI Ranges | Build-Up Area Classification | Class |
|---|---|---|
| −1.0 to 0.0 | Non-Built-up areas | 1 |
| 0.0 to +0.1 | Built-up areas | 2 |

### 2.3. Methodology

The workflow is divided into the following parts: (a) Meteorological Data and Land Surface Temperature Evaluation Methods, (b) LULC and LST Comparative and Correlation Analysis, (c) LST Spatiotemporal Pattern Analysis and Hotspots/Cold spots Identification, and (d) Intra-Urban Heat Island Map Generation.

The overall workflow of this methodology is shown in Figure 2. Finally, using the information obtained, data assessment and suggested area-specific mitigation strategies are provided.

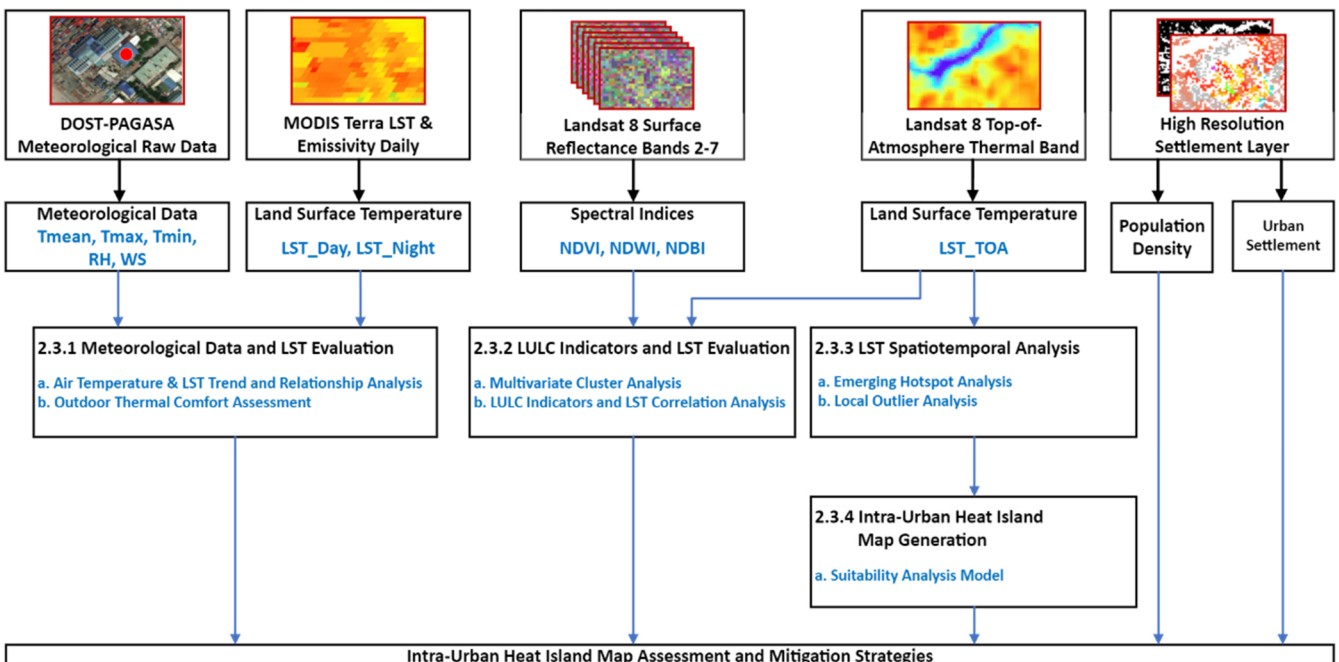

**Figure 2.** Overview of the overall workflow of the study to assess the IUHI map and provide mitigation strategies.

2.3.1. Meteorological Data and Land Surface Temperature Evaluation Methods

This section focuses on the use of meteorological data collected at Port Area, Manila City, and how they are used to understand the temporal variability of air temperature, the relationship of meteorological parameters to land surface temperature during the day and night, and outdoor thermal comfort assessment.

a.      Air Temperature and LST Trend and Relationship Analysis

This analysis's methodology and findings were already published by the authors in ref. [45]. There was no gap-filling technique used for missing information related to the in-situ measurements nor with the derived MODIS data specific to the meteorological data point. The in-situ data were directly taken from the weather agency which processed and prepared the data, while the MODIS data are directly downloaded from the Google earth engine. All data used were analysis-ready while any data point with a missing parameter entry was discarded and not used.

b.      Outdoor Thermal Comfort Assessment

The RayMan Model was proposed by Matzarakis, a micro-scale model developed to calculate radiation fluxes in simple and complex environments [55,56]. This research used this model to assess the thermal comfort in Port Area. The scientific basis for the computations is thoroughly detailed in the Rayman Pro tool handbook [55].

Thermal indices have been developed to approximate human thermal perception [55]. In particular, Physiological Equivalent Temperature (PET) is "the air temperature at which, in a typical indoor setting (without wind and solar radiation), the energy budget of the human body is balanced with the same core and skin temperature as under the complex outdoor conditions to be assessed" [57,58].

The Thermal Comfort Assessment workflow is as follows:

1.      Preparation of input parameters (Air Temperature, Relative Humidity, and Wind Velocity) in a .csv file as input to the RayMan Model.
2.      Calculate the Tmrt and Thermal Index (PET) using the RayMan Pro Software. The Graphical User Interface which contains the geographic data, personal data, and clothing & activity information used is shown in Figure 3.
3.      Graph the calculated values for comparison.
4.      Assess the thermal comfort by getting the equivalent physiological stress associated with the derived thermal index values as shown in Table 5.

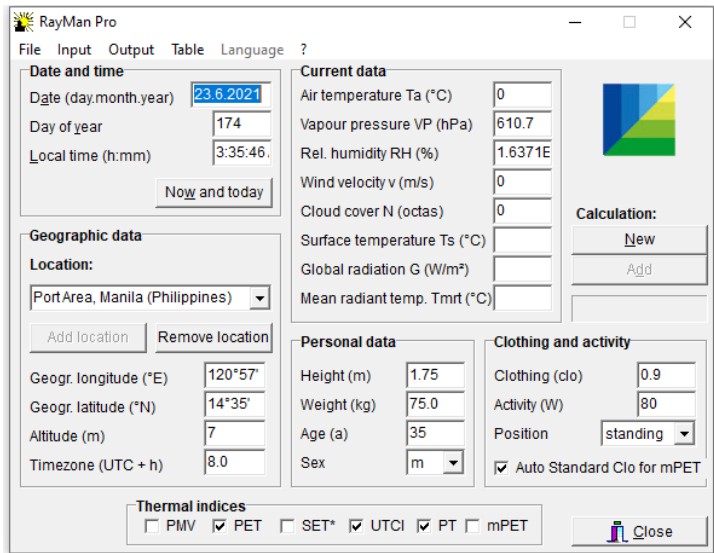

**Figure 3.** RayMan Pro Graphical User Interface. Geographic data, personal data, and clothing and activity information are shown.

**Table 5.** PET Thermal Index, corresponding classes, thermal sensation, and physiological stress.

| Thermal Sensation | PET Range for Taiwan (°C PET) [59] | PET Range for Western/Middle Europe (°C PET) [59] | Physiological Stress |
|---|---|---|---|
| Very Cold | <+14 | <+4 | Extreme cold stress |
| – | – | – | Very strong cold stress |
| Cold | +14–+18 | +4–+8 | Strong cold stress |
| Cool | +18–+22 | +8–+13 | Moderate cold stress |
| Slightly Cool | +22–+26 | +13–+18 | Light cold stress |
| Neutral | +26–+30 | +18–+23 | No thermal stress (Thermal Comfort Zone) |
| Slightly Warm | +30–+34 | +23–+29 | Light heat stress |
| Warm | +34–+38 | +29–+35 | Moderate heat stress |
| Hot | +38–+42 | +35–+41 | Strong heat stress |
| – | – | – | Very strong heat stress |
| Very Warm | >+42 | >+41 | Extreme heat stress |

It should be emphasized that the data being used in this analysis are solely temporal point data from Manila City's Port Area. It is deemed that these values do not represent the entire city; therefore, meteorological data-point locations should be explored to offer a better understanding of the thermal comfort in Manila City.

### 2.3.2. LULC Indicators and LST Evaluation Methods

This section discusses methods to evaluate satellite-derived data such as spectral indices (NDVI, NDWI, and NDBI, which are used as LULC indicators) and land surface temperature in Manila City. These methods include multivariate cluster analysis and correlation analysis.

a. Multivariate Cluster Analysis

Cluster analysis is a statistical method to use the values of the variables in devising a scheme for grouping the objects into classes so that similar objects are in the same class [60]. It is a multivariate method for classifying a sample of subjects (or objects) into several groups based on a set of measured characteristics, with related subjects placed in the same group.

Given that the group of values for each parameter is not known, we used the satellite-derived data to group the values in each parameter (NDVI, NDWI, NDBI) together with land surface temperature (LST) and observed how each of these LULC indicators relate to LST. Specifically, since the indicator values can be used to classify land use and land cover, this is an initial step to see how the land use and land cover of different areas in Manila City relate to their thermal characteristic.

For this, the k-means algorithm as shown in Algorithm 1 was used to identify the clusters within the dataset. It is an iterative algorithm that divides the unlabeled dataset into k different clusters in such a way that each dataset belongs to only one group that has similar properties [61]. The k-means clustering algorithm mainly performs two tasks: (1) determine the best value for k-center points or centroids by an iterative process and (2) assign each data point to its closest k-center. Those data points which are near a particular k-center create a cluster. Hence, each cluster has data points with some commonalities, and it is away from other clusters. Shown below is the k-means clustering algorithm flow.

| **Algorithm 1:** k-means algorithm |
|---|
| 1: Specify the number k of clusters to assign. |
| 2: Randomly initialize k centroids. |
| 3: Repeat |
| 4:     expectation: Assign each point to its closest centroid |
| 5:     maximization: Compute the new centroid (mean) of each cluster. |
| 6: until the centroid positions do not change. |

In this study, we used the multivariate clustering tool in ArcGIS Pro [62] to find these natural clusters of features based solely on the feature attribute values. Given the number of clusters to create, it will look for a solution where all the features within each cluster are as similar as possible, and all the clusters themselves are as different as possible. This tool utilizes unsupervised machine learning methods to determine natural clusters in the data. The classification method is considered unsupervised as they do not require a set of reclassified features to guide or train the method to find the clusters in the data. Since the tool is used to run the clustering algorithm, the following workflow was employed:

1.  Extract the values from the raster map at different years to create a feature layer. The spectral indices (NDVI, NDWI, & NDBI) are in values between −1 and 1 while land surface temperature is in degrees Celsius (°C). All the raster data are taken from Landsat 8 as explained in Section 2.2.4.
2.  Import the data into the ArcGIS Pro software and use the generated feature layer as input.
3.  Execute the k-means clustering algorithm with the following:
4.  Clustering method: k-means
5.  Initialization Method: Optimized seed locations
6.  Number of clusters: 4
7.  Generate the cluster chart and interpret the results according to each of the input variables.

It should be noted that cluster analysis has no mechanism for differentiating between relevant and irrelevant variables. Therefore, the choice of variables included in a cluster analysis must be underpinned by conceptual considerations. This is very important because the clusters formed can be very dependent on the variables included. To see the relationship and extent of the values used in clustering, we also employed correlation analysis with the data.

b.　LULC Indicators and LST Correlation Analysis

We use correlation analysis in addition to multivariate clustering analysis to evaluate the relationship of NDVI, NDWI, and NDBI with LST. The same method as explained in Section 2.3.1-a was used to analyze the extent and nature of the relationship between the abovementioned parameters. On the contrary, Pearson product correlation in GeoDa software was used.

### 2.3.3. LST Spatiotemporal Pattern Analysis

In this section, we focus on analyzing the spatial and temporal pattern of Land Surface Temperature in Manila City Philippines. Since data have both spatial and temporal context, several analytical tools in the Space-Time Pattern Analysis toolset in ArcGIS Pro software [62] were used. Before doing the analysis, a space-time cube was created based on the downloaded LST raster over the period (2014 to 2021) as shown in Figure 4.

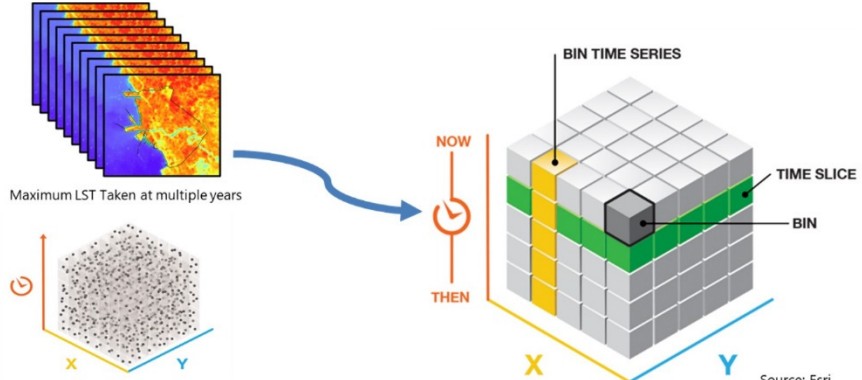

**Figure 4.** Creating a space-time cube based on yearly maximum LST from 2013 to 2022.

A time series analysis or an integrated spatial and temporal pattern analysis may be used to view and analyze spatial-temporal data using this approach. Using the prepared space-time cube as input, we perform emerging hotspot analysis and local outlier analysis to better understand the thermal situation in Manila City.

a.　　Emerging Hotspot Analysis

The Emerging Hot Spot Analysis tool detects statistically significant hot and cold spot patterns over time. It is used to examine land surface temperature (LST) data in Manila City to identify new, intensifying, persistent, or sporadic hot spot trends at various time-step intervals. The workflow for this is as follows:

1. Taking the space-time NetCDF cube created for LST as input.
2. Conceptualize the spatial relationships of LST values using the k-nearest neighbor method with k = 8, where the eight closest neighbors to the target feature will be included in computations for that feature.
3. Calculate the Getis-Ord *Gi\** statistic [63] for each bin (pixel), represented in Table 6. The Getis-Ord local statistic is given as:

$$G_i^* = \frac{\sum_{j=1}^{n} w_{i,j} x_j - \overline{X} \sum_{j=1}^{n} w_{i,j}}{S \sqrt{\frac{\left[ n \sum_{j=1}^{n} w_{i,j}^2 - \left( \sum_{j=1}^{n} w_{i,j} \right)^2 \right]}{n-1}}} \tag{4}$$

where $x_j$ is the attribute value for feature $j$, $w_{i,j}$ is the subscript weight between feature $i$ and $j$, $n$ is equal to the number of features; also:

$$\overline{X} = \frac{\sum_{j=1}^{n} x_j}{n} \tag{5}$$

$$S = \sqrt{\frac{\sum_{j=1}^{n} x_j^2}{n} - \left( \overline{X} \right)^2} \tag{6}$$

The $G_i^*$ is a z-score so no further calculations are required.

**Table 6.** $G_i^*$ statistic values for cold spot and hotspot classes at different significance levels.

| Statistical Significance Level | $G_i^*$ Statistic Pixel Representation | |
| --- | --- | --- |
| | **Cold Spot** | **Hotspot** |
| 99% confidence | −3 | +3 |
| 95% confidence | −2 | +2 |
| 90% confidence | −1 | +1 |
| Statistically not significant | 0 | |

The $G_i^*$ statistic returned for each point is a z-score. The more concentrated the clustering of high values (hot spots) of LST, the bigger the z-score for statistically significant positive z-scores. The clustering of low values (called a "cold spot") of LST is stronger, the smaller the z-score is for statistically significant negative z-scores.

4. Once the space-time hot spot analysis completes, each bin (pixel) in the input NetCDF cube has an associated z-score, *p*-value, and hot spot bin classification added to it.
5. Next, these hot and cold spot trends are evaluated using the Mann–Kendall trend test. As an independent bin time-series test, the Mann–Kendall trend test [64] is done for every location/point with LST data. For the point value and their time sequence, the Mann–Kendall statistic is a rank correlation analysis. The first time's point value is compared to the second time's point value. The outcome is +1 if the first is smaller than the second. The outcome is −1 if the first is greater than the second. The outcome is 0 if the two numbers are equal. The results are added together for each pair of periods compared. The predicted sum is 0, indicating that the numbers do not show

any trend over time. Based on the variance for the values in the point time series, the number of ties, and the number of periods, the observed sum is compared to the expected sum (zero) to determine if the difference is statistically significant. A z-score and a *p*-value are used to represent the trend for each point time series. A small *p*-value indicates that the trend is statistically significant. The sign associated with the z-score determines if the trend is an increase in point values (positive z-score) or a decrease in bin values (negative z-score).

With the resultant trend z-score and *p*-value for each location with data, and with the hot spot z-score and *p*-value for each bin, the Emerging Hot Spot Analysis tool in ArcGIS Pro categorizes each study area location as shown in Table 7 and is then reclassified as "monitor", "intervene", and "preserve". With the new classification, those categorized as diminishing, oscillating, and historical for both hot and cold spots will be reclassified as "monitor". Those with no pattern detected will be classified as "monitor" as well. On the other hand, categories such as new, consecutive, intensifying, and sporadic will have "preserve" as their new class for a cold spot and "intervene" for a hotspot.

6.  An Emerging Hotspot Analysis (EHSA) Map showing areas to preserve, monitor, and intervene is generated based on the reclassification shown in Table 7.

**Table 7.** Emerging hot spot analysis trend categories, their definition, and equivalent new class.

| Category | | Definition | New Class |
|---|---|---|---|
| No Pattern Detected | | Does not fall into any of the hot or cold spot patterns defined below | Monitor |
| Hot Spot | New | the most recent time step interval is hot for the first time | Intervene |
| | Consecutive | a single uninterrupted run of hot time step intervals, with of less than 90% of all intervals | Intervene |
| | Intensifying | at least 90% of the time step intervals are hot and become hotter over time | Intervene |
| | Persistent | at least 90% of the time step intervals are hot, with no trend up or down | Intervene |
| | Sporadic | some of the time step intervals are hot | Intervene |
| | Diminishing | at least 90% of the time step intervals are hot and become less hot over time | Monitor |
| | Oscillating | some of the time step intervals are hot, some are cold | Monitor |
| | Historical | at least 90% of the time step intervals are hot, but the most recent time step interval is not | Monitor |
| Cold Spot | New | the most recent time step interval is cold for the first time | Preserve |
| | Consecutive | a single uninterrupted run of cold time step intervals, withof less than 90% of all | Preserve |
| | Intensifying | at least 90% of the time step intervals are cold and become colder over time | Preserve |
| | Persistent | at least 90% of the time step intervals are cold, with no trend up or down | Preserve |
| | Sporadic | some of the time step intervals are cold | Preserve |
| | Diminishing | at least 90% of the time step intervals are cold and become less cold over time intervals | Monitor |
| | Oscillating | some of the time step intervals are cold, some are hot | Monitor |
| | Historical | at least 90% of the time step intervals are cold, but the most recent time step interval is not | Monitor |

b.  Local Outlier Analysis

The Local Outlier Analysis tool identifies statistically significant clusters of high or low land surface temperature LST values as well as outliers that have values that are statistically different from their neighbors in space and time.

The workflow for this is as follows:

1.  Use the space-time NetCDF cube created for LST as input.
2.  Conceptualize the spatial relationships of LST values using the k-nearest neighbor method with k = 8, where the eight closest neighbors to the target feature will be included in computations for that feature.

3. Calculate the Anselin Local Moran's *I* statistic of special association for each bin which includes a pseudo *p*-value and a CO_Type code. The Local Moran's *I* statistic of spatial association is given as

$$I_i = \frac{x_i - \overline{X}}{S_i^2} \sum_{j=1,\, j \neq i}^{n} w_{i,j} \left( x_i - \overline{X} \right) \tag{7}$$

where $x_i$ is an attribute for feature *i*, $\overline{X}$ is the mean corresponding attribute, $w_{i,j}$ is the spatial weight between features *i* and *j*, and:

$$S_i^2 = \frac{\sum_{j=1,\, j \neq i}^{n} \left( x_j - \overline{X} \right)^2}{n-1} \tag{8}$$

with *n* equating to the total number of features. The $z_{I_i}$ score for the statistics is computed as

$$z_{I_i} = \frac{I_i - E[I_i]}{\sqrt{V[I_i]}} \tag{9}$$

$$V[I_i] = E\left[I_i^2\right] - E[I_i]^2 \tag{10}$$

A positive value for *I* indicates that a feature has neighboring features with similarly high or low attribute values; this feature is part of a cluster. A negative value for I indicates that a feature has neighboring features with dissimilar values; this feature is an outlier. In either instance, the *p*-value for the feature must be small enough for the cluster or outlier to be considered statistically significant.

In Table 8, the cluster/outlier type (CO Type) field distinguishes between a statistically significant cluster of high values (HH), a cluster of low values (LL), an outlier in which a high value is surrounded primarily by low values (HL), and an outlier in which a low value is surrounded primarily by high values (LH). Statistical significance is set at the 95 percent confidence level. This significance represents an FDR correction, which adjusts the *p*-value threshold from 0.05 to a value that better reflects the 95 percent confidence level taking into consideration multiple testing.

4. A two-dimensional map summarizing each location over time is created with the following categories shown in Table 9. Then, a new class is created based on these categories wherein pixels categorized as never significant, multiple types and outliers will be reclassified as "monitor" while only the high-high cluster and the low-low cluster will be reclassified as intervene and preserve, respectively.

5. Finally, a Local Outlier Analysis (LOA) Map showing areas to preserve, monitor, and intervene will be generated.

**Table 8.** Pixel representation of cluster and outliers based on the Anselin Local Moran's *I* statistic.

| Cluster/Outlier Type | Definition |
|---|---|
| Never Significant | A location that is not statistically significant. |
| High-High Cluster (HH) | Locations that are part of a cluster of high LST_TOA values. |
| High-Low Outlier (HL) | Locations that represent high outliers within a cluster of low LST_TOA values. |
| Low-High Outlier (LH) | Locations that represent low outliers within a cluster of high LST_TOA values. |
| Low-Low Cluster (LL) | Locations that are part of a cluster of low LST_TOA values. |

**Table 9.** Local outlier analysis trend categories, their definition, and equivalent new class.

| Category | Definition | New Class |
|---|---|---|
| Never Significant | A location where there has never been a statistically significant CO_TYPE. | Monitor |
| Only High-High Cluster | A location where the only statistically significant type throughout time has been High-High Clusters. | Intervene |
| Only High-Low Outlier | A location where the only statistically significant type throughout time has been High-Low Outliers. | Monitor |
| Only Low-High Outlier | A location where the only statistically significant type throughout time has been Low-High Outliers. | Monitor |
| Only Low-Low Cluster | A location where the only statistically significant type throughout time has been Low-Low Clusters. | Preserve |
| Multiple Types | A location where there have been multiple types of statistically significant clusters and outlier types throughout time. | Monitor |

### 2.3.4. Intra-Urban Heat Island Map Generation

This section discusses the method of generating the intra-urban island map for Manila City, Philippines, using results from EHSA and LOA through a Suitability Analysis Model.

Figure 5 shows the overall process to produce the needed map for further assessment. The Emerging Hot Spot Analysis identifies trends in the data, such as new, intensifying, diminishing, and sporadic hot and cold spots, while the Local Outlier Analysis identifies significant clusters and outliers in the data. Through the suitability analysis, the combination of both methods ensures that locations of hot and cold spots in the city are precisely identified by eliminating outlier clusters in the final map produced. The suitability analysis model was used to combine the resulting raster map from the emerging hotspot analysis and local outlier analysis.

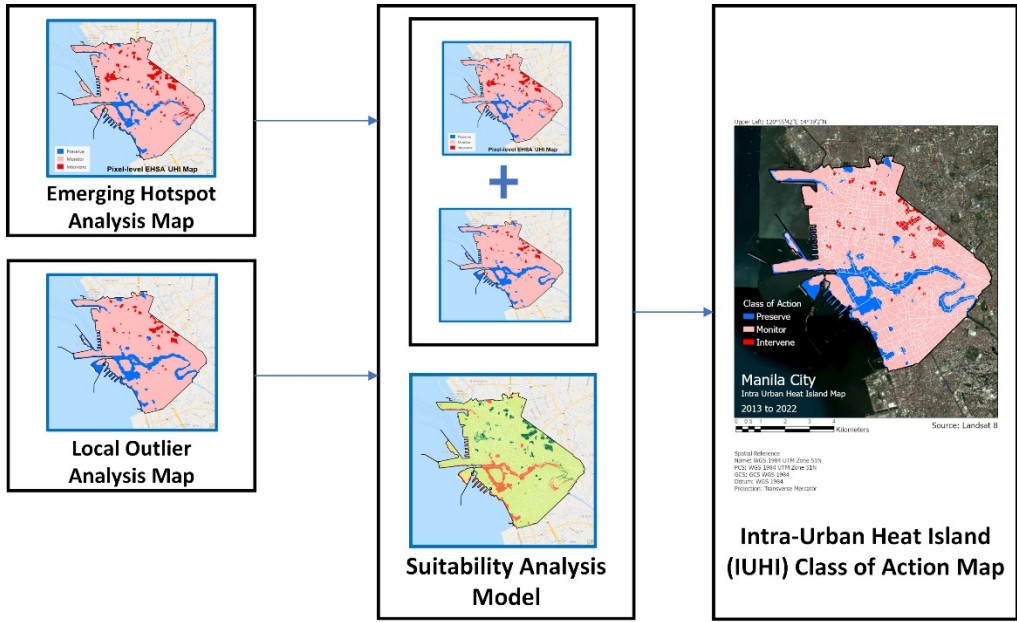

**Figure 5.** Overview of the Intra-Urban Heat Island (IUHI) Class of Action map generation based on EHSA and LOA maps using the suitability analysis model.

To carry out the suitability analysis, the classification classes of emerging hotspot analysis and local outlier analysis were given numerical equivalents to provide a common suitability scale.

Specifically, the following workflow was followed:

1.  Preparation of criteria data. The resulting maps from the emerging hotspot analysis and local outlier analysis were prepared with their corresponding classes.
2.  Transforming the classes of each criterion to a common suitability scale is shown in Table 10.
3.  Assigning weight relative to each of the criteria and combining them to create a suitability map. In this application, we treat each criterion as equally important, so weight is assigned as a percentage: 50% for EHSA Classification and 50% for LOA Classification.
4.  Finally, the pixel values were reclassified according to Table 11, shown to give an Intra-Urban Heat Island (IUHI) Class of Action Map.

**Table 10.** Common suitability scale used to transform EHSA and LOA Classification maps.

| Emerging Hotspot Analysis (EHSA) Classification | Local Outlier Analysis (LOA) Classification | Suitability Scale |
|---|---|---|
| Preserve | Preserve | 1 |
| Monitor | Monitor | 2 |
| Intervene | Intervene | 3 |

**Table 11.** Suitability values and their equivalent IUHI Class of Action.

| Emerging Hotspot Analysis (EHSA) Classification | Local Outlier Analysis (LOA) Classification | Suitability Model Suitability Value | IUHI Class of Action |
|---|---|---|---|
| 1 | 1 | 1.0 | Preserve |
| 1 | 2 | 1.5 | Preserve |
| 2 | 1 | 1.5 | Preserve |
| 1 | 3 | 2.0 | Monitor |
| 2 | 2 | 2.0 | Monitor |
| 3 | 1 | 2.0 | Monitor |
| 2 | 3 | 2.5 | Monitor |
| 3 | 2 | 2.5 | Monitor |
| 3 | 3 | 3.0 | Intervene |

### 2.3.5. Intra-Urban Heat Island Map Assessment and Mitigation Strategies

The results in Sections 2.3.1–2.3.4 are then used to evaluate the Intra-Urban Heat Island map with the population data and urban settlement raster from the high-resolution settlement layer. Moreover, area-specific mitigation strategies will be suggested based on the visual inspection of the areas that need intervention. Possible strategies may also be taken from the identified areas to be preserved in the city. Assessment and mitigation strategies are simplified so that they serve as a basis for urban planners and policymakers for mitigation and improvement.

## 3. Results

### 3.1. Satellite Data Retrieved from Landsat 8

Ten distribution maps from 2013 to 2022 were obtained from Landsat 8 data through the climate engine web application. These data were further processed in ArcGIS Pro by providing an equalized histogram stretch and a specific color scheme in its symbology.

### 3.2. Meteorological Data and Land Surface Temperature Evaluation

3.2.1. Air Temperature and LST Trend and Relationship Analysis

Figure 6 shows the monthly maximum (Tmax), mean (Tmean), and minimum (Tmin) air temperature trends from 2014 to 2018. The values were taken from the diurnal data and were averaged per month to clearly show the monthly trend. This observation was discussed in [45] showing an upward trend in the values starting from March and continuing to April and May while values start to drop in October until around January and February. Such an observation is the same as what was presented by Estoque et al. [28] and Manalo et al. [65] in their framework showing the climate and seasons in the Philippines based on combined rainfall and temperature. Between March to May, the Philippines experiences a hot dry season which explains the high recorded air temperature.

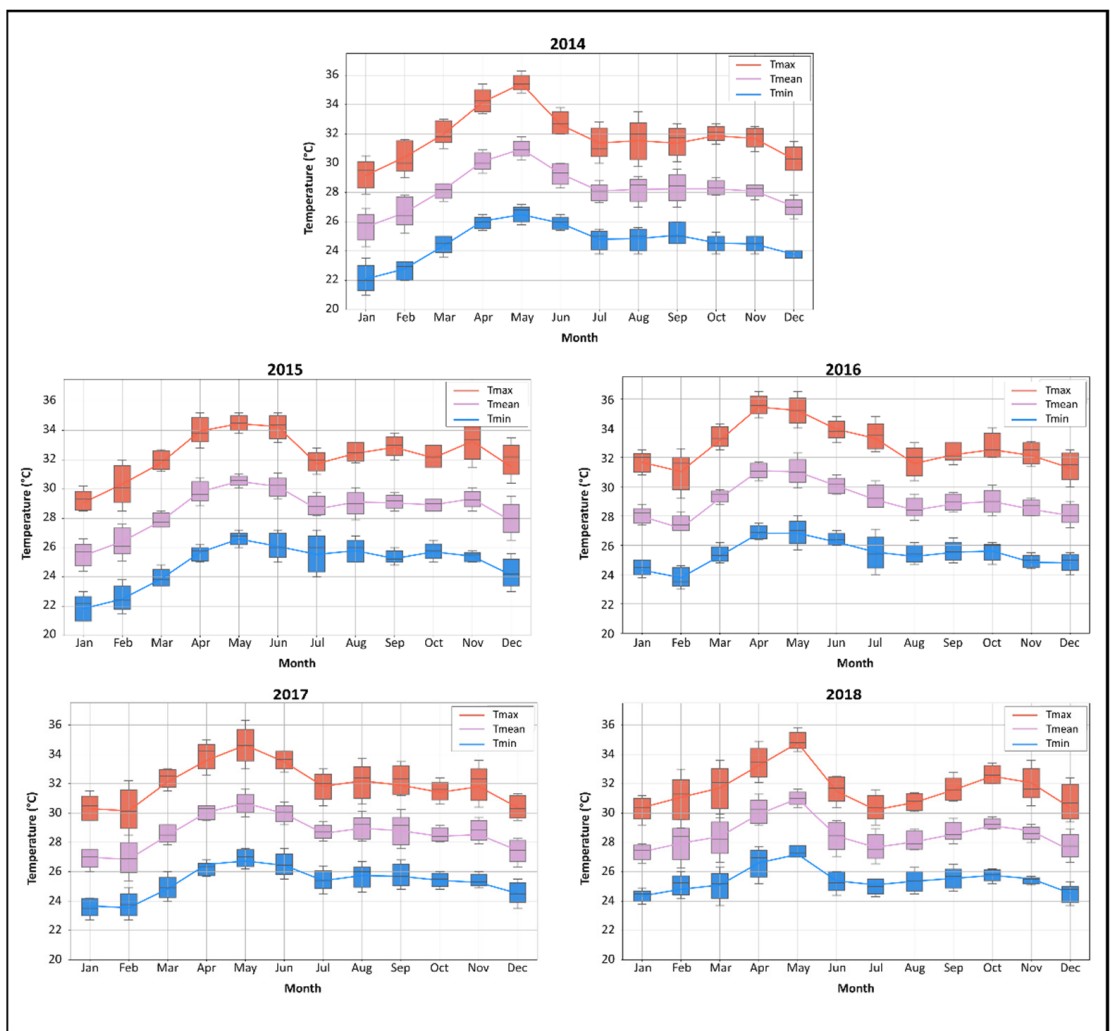

**Figure 6.** Monthly maximum (Tmax), mean (Tmean), and minimum (Tmin) air temperature trends from 2014 to 2018.

Additionally in our paper [45], we found a significant linear correlation between air temperature (maximum, mean, and minimum) and land surface temperature (day and night) as analyzed from available daily data shown in Table 12. On the other hand, the relative humidity shows a weak correlation with the LST data although it is shown to be significant for LST_Night.

**Table 12.** Corresponding interpretation of the quantitative values from the correlation analysis [45]. (* not significant).

|  | LST_Day | LST_Night |
|---|---|---|
| Tmax | moderate | strong |
| Tmean | moderate | strong |
| Tmin | moderate | strong |
| RH | weak * | weak |

### 3.2.2. Outdoor Thermal Comfort Assessment

Using the same meteorological data (Tmean, Relative Humidity, and Wind Speed) taken in Port Area, Manila City, from 2014 to 2018, the Physiological Equivalent Temperature (PET) thermal index was estimated through the RayMan model. The diurnal data were computed and then averaged per month and are shown in Figure 7. Additionally, the corresponding physiological stress levels for each of the values are indicated.

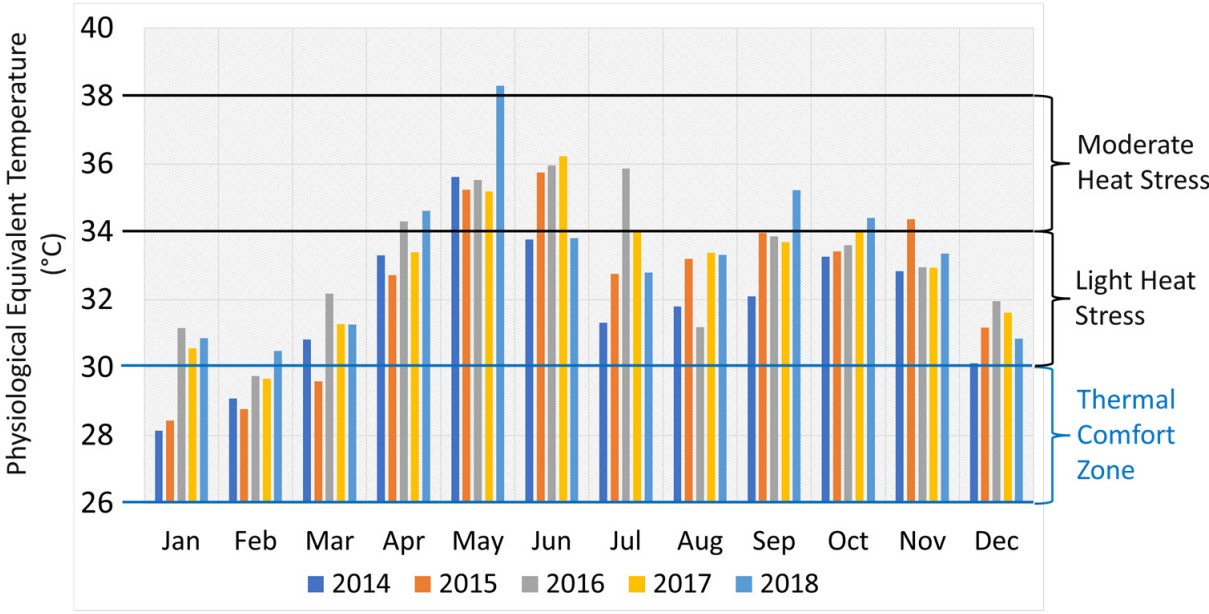

**Figure 7.** Monthly estimated Physiological Equivalent Temperature (PET) based on the RayMan model from 2014 to 2018.

As shown, moderate heat stress can be consistently felt in May and at some points in April and June. From July to December, light heat stress was observed, while the thermal comfort zone where there is no thermal stress only appeared in January and February. Understanding the thermal comfort in this area can also give us an idea on what is the expected outdoor thermal comfort in the other parts of Manila City. These results will be used as part of the assessment method in the latter part of the study.

### 3.3. LULC Indicators and Evaluation Methods
### 3.3.1. Multivariate Cluster Analysis

From the space-time cube generated for spectral indices (NDVI, NDWI, and NDBI) used as land use and land cover indicators and top-of-atmosphere land surface temperature (TOA_LST), the k-means clustering algorithm was used to identify the clusters within the dataset. Four groups were initialized to see a cluster for high LST (1 cluster), mid-LST (2 clusters), and low LST values (1 cluster). Standardized parameter values were plotted to clearly show the distribution of clusters, as the measurement units are not the same.

Figure 8 shows the boxplot of the result of the multivariate cluster analysis. The clustering results indicate that for the high LST cluster, values with low NDWI, moderate NDVI, and high NDBI values are clustered together. This is also expected since low NDWI correlates to low water content and high NDBI corresponds to urbanized regions. In contrast, mid-range NDVI values correspond to urbanized areas. For the low LST cluster, values are clustered with high NDWI values, low NDVI values, and low NDBI values. A high NDWI refers to a high-water content, a negative NDVI to water bodies, and a low NDBI to undeveloped regions. Consequently, two mid-LST clusters were produced because of varying parameter combinations. The first set of clusters for mid-LST (orange line) is seen to be a combination of negative NDBI, high NDVI, and a higher mid-value of NDWI which translates to lowly built-up, high vegetation with a fair amount of moisture content. On the other hand, the second set of mid-LST clusters (light blue line) is composed of NDBI, NDVI, and NDWI values close to zero which can be interpreted as areas with low to no built-up and low water content.

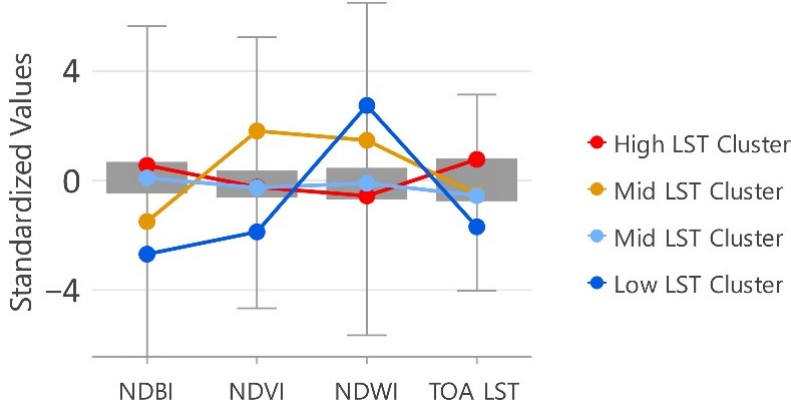

**Figure 8.** Boxplot of the multivariate cluster analysis result.

3.3.2. LULC Indicators and LST Correlation Analysis

The same dataset was used to see the correlation of these parameters (NDVI, NDWI, NDBI) with land surface temperature (TOA_LST). GeoDa software was used to calculate the Pearson correlation and plot the results.

Figure 9 shows the relationship between LST and LULC indicators with their corresponding slope of linear fit and frequency distribution chart while all indicators are significant at $p < 0.01$. The results show that there is a direct relationship between LST and NDBI at a $r = 0.361$ which means that highly built-up areas have high recorded temperature values. This observation agrees with the multivariate analysis. An indirect relationship is, however, observed between LST and NDVI ($r = -0.064$) and LST and NDWI ($r = -0.365$). The low Pearson correlation value between LST and NDVI indicates that both water body values and vegetation are expected to have low temperatures while mid values correspond to being built-up. With LST and NDWI, areas with high water/moisture content are more likely to have lower surface temperatures compared to areas with low water/moisture content. Based on these results, it can be inferred that the correlation values suggest that NDWI is a better indicator than NDVI for land surface temperature, which is aligned with the findings of Alexander et al. [66]. In addition, results also suggest that NDBI is a good indicator for LST.

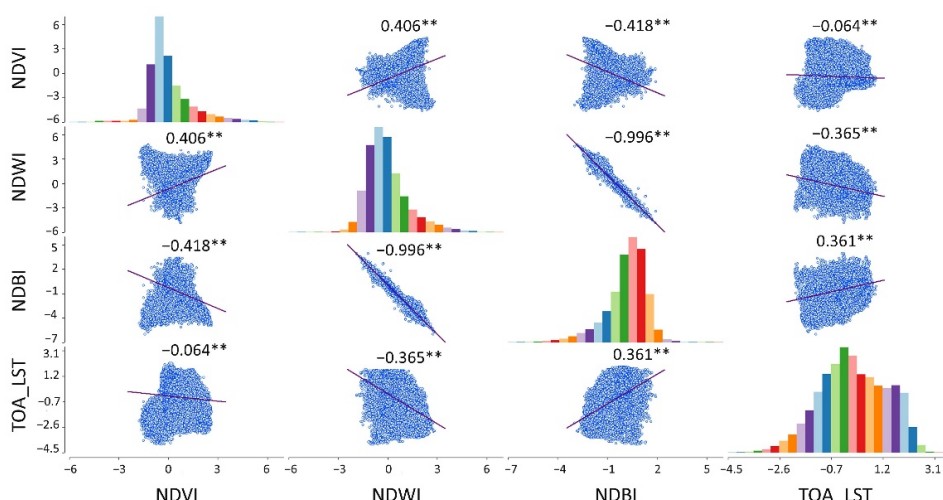

**Figure 9.** Relationship between LST and spectral indices with their corresponding slope of linear fit and frequency distribution chart. ** significant at $p < 0.01$.

### 3.4. LST Spatiotemporal Pattern Analysis

3.4.1. Emerging Hotspot Analysis

Based on the generated Emerging Hotspot Analysis (ESHA) Map, a reclassified map was also produced to indicate areas to preserve, monitor, and intervene.

As shown in Figure 10, cold spot and hot spot areas were mapped using the trend categories and a corresponding new class.

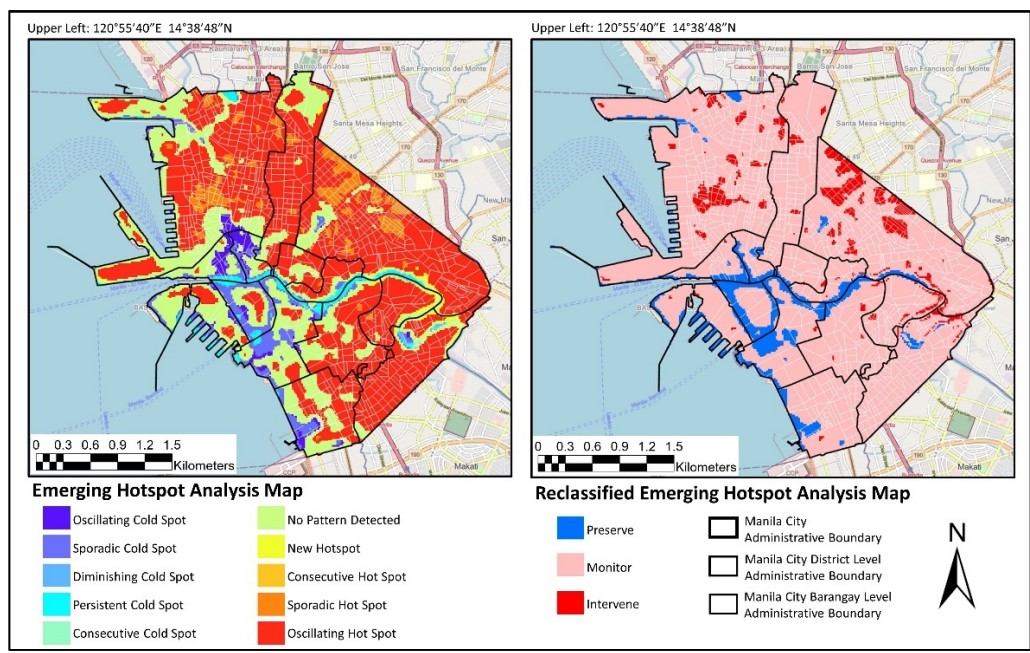

**Figure 10.** Emerging Hotspot Analysis Map and the reclassified map with the corresponding new class.

3.4.2. Local Outlier Analysis

Based on the generated Local Outlier Analysis (LOA) Map, a reclassified map was also produced to indicate areas to "preserve", "monitor", and "intervene". In Figure 11, the trend categories of clusters and outliers are shown on the left while the corresponding new class is also provided in the map on the right.

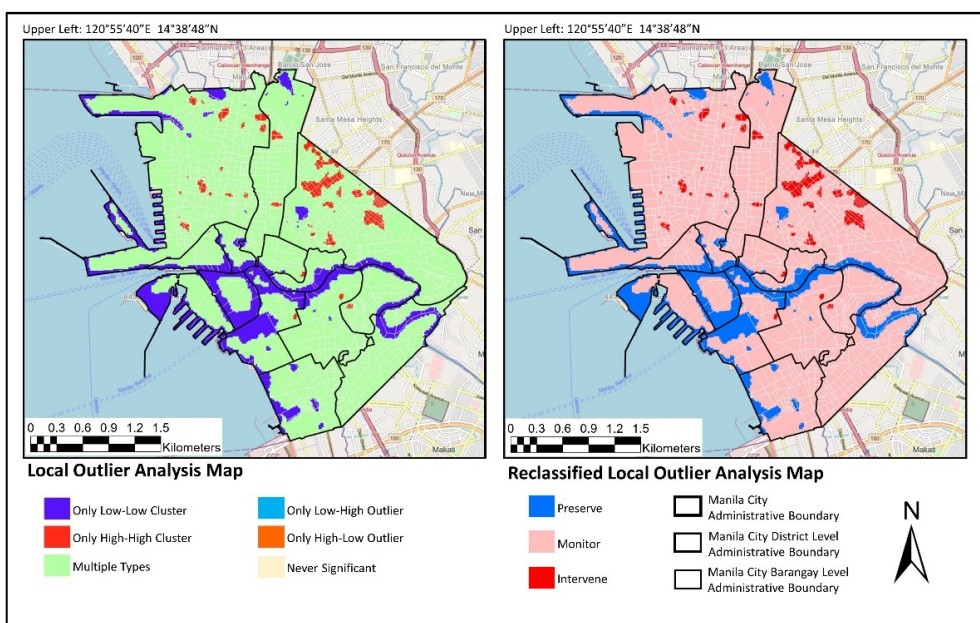

**Figure 11.** Local Outlier Analysis Map and the reclassified map with the corresponding new class.

*3.5. Intra-Urban Island Map*

Using the generated maps presented in Sections 3.2.1 and 3.2.2, a suitability analysis model was used to combine the raster maps. The suitability analysis was carried out by giving numerical equivalents for the new classification maps for emerging hotspot analysis and local outlier analysis with a common suitability scale.

Figure 12 (left) shows the resulting suitability map with suitability values per pixel. Consequently, the equivalent Intra-Urban Heat Island (IUHI) Class of Action was produced as shown in Figure 12 (right).

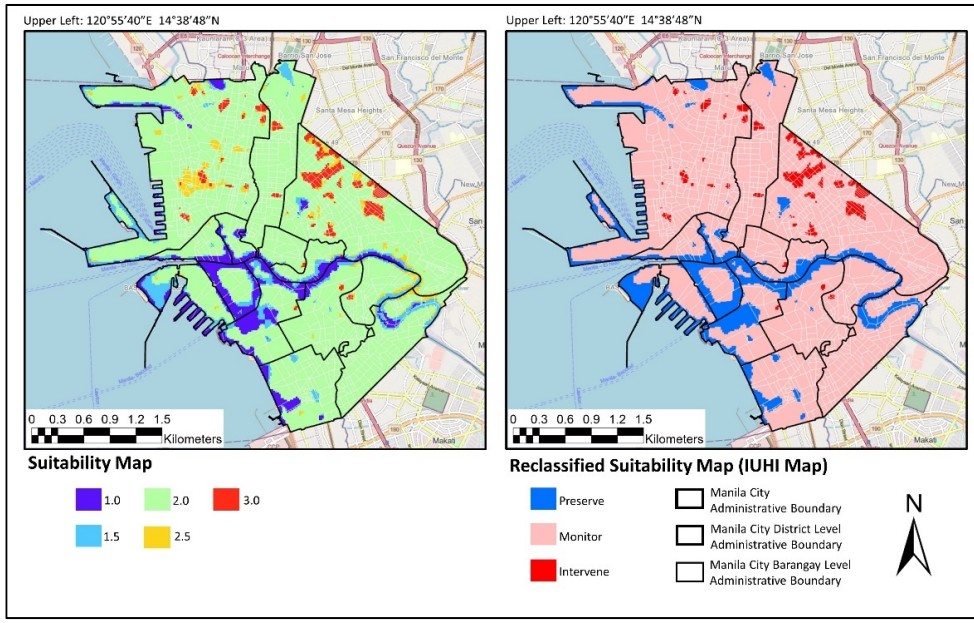

**Figure 12.** Suitability Map and the reclassified suitability (IUHI) map with the corresponding new class.

In Figure 13, the final Intra-Urban Heat Island (IUHI) Map of Manila City (2013–2022) was created. To keep the map as intuitive as possible, the class of action as well as the administrative boundaries at the city, district, and barangay levels were provided. This

allows an easy understanding of the map while still showing the locations where areas need preservation, monitoring, and intervention.

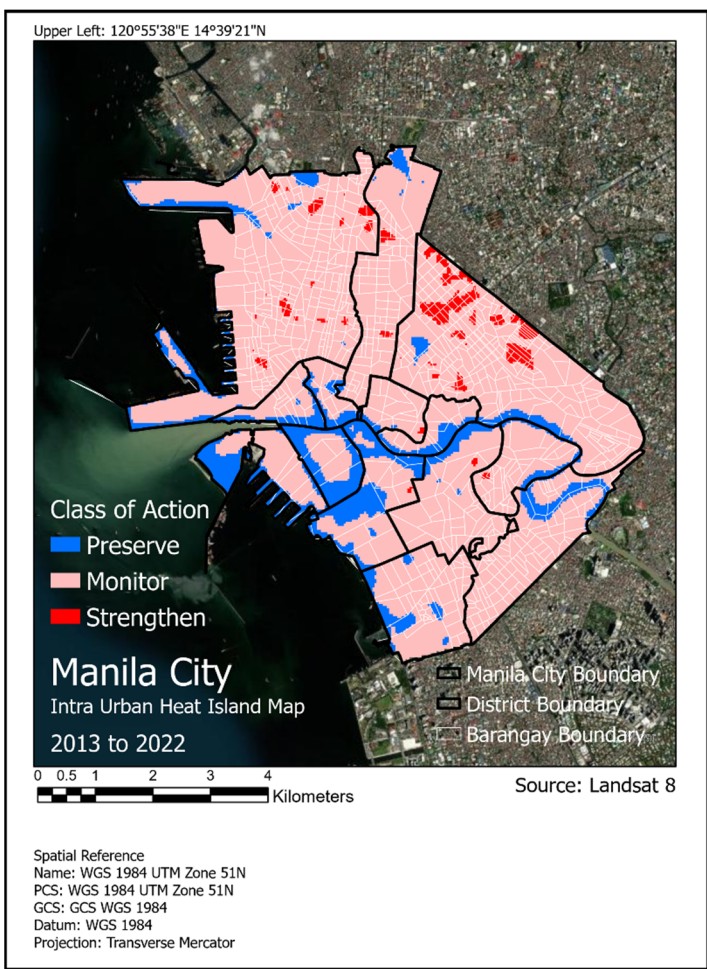

**Figure 13.** Intra-Urban Heat Island Map of Manila City (2013–2022).

*3.6. Intra-Urban Heat Island Map Assessment and Mitigation Strategies*

3.6.1. Location Assessment

Using the IUHI Map of Manila City, areas classified as "preserve" and "intervene" were examined visually using high-resolution maps from Google Earth Pro.

From the IUHI map, areas that need intervention were assessed by visually inspecting the locations to see the morphology of the areas exhibiting consistent surface temperatures during the study period. Based on the inspection, most of these areas fall within the Sampaloc district, which is part of Manila City's university belt shown in Figure 14E–H catering to Manila's academic population. The area's abundance of hotels and boarding houses makes it ideal as a dormitory and as a commuting town [36]. Moreover, there are also a few areas situated in Tondo District (A, B, and C) which is among the biggest urban poor communities in Manila City. Area D, on the other hand, mainly points toward a commercial location in Paco District.

Looking at the high-resolution satellite images, the areas shown in Figure 14 represent commonality in terms of their urban structure. It is noticeable that these areas (A, B, C, E, F, G, and H) are mostly residential and is characterized by predominantly settlement and housing locations with narrow streets and sidewalks. Although there are attempts to introduce urban soft scape via trees and vegetation, these are few and sparsely distributed within the areas of concern. In general, roads and walkways are mainly built with asphalt and concrete which might contribute to higher surface temperatures. There is also commer-

cial space identified, such as (D), which seemed to have establishments and buildings and parking spaces made of either asphalt or concrete as well.

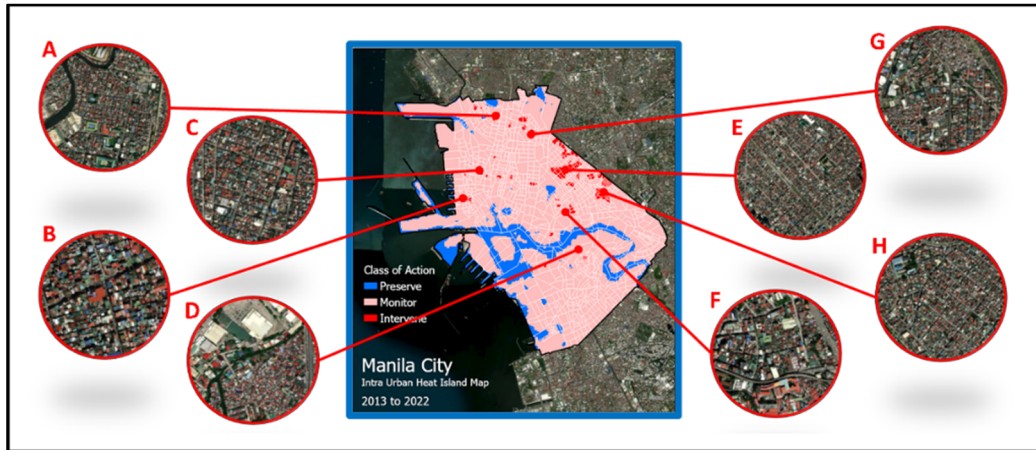

**Figure 14.** Some areas with the "Intervene" Class of Action. (**A–H**) are the areas highlighted to show their morphologies.

The same approach was applied in examining the areas to be preserved shown in Figure 15. Aside from the stretch of Pasig River amidst Manila City, the Intramuros district including Rizal Park Complex (part of Ermita district) as shown in (D) shows large areas with relatively lower surface temperatures. It is the historic core of Manila and is described as the "walled city" where walls surrounding the area are present until today. The Intramuros area has evident low surface temperature due to its strategic location. Aside from being situated near a body of water (Pasig River), the area is surrounded by greenery (mostly grass and some shrubs and trees) which is part of a golf range. On the other hand, the Rizal Park complex is one of the largest urban parks in Asia wherein the area is a combination of vegetation and trees, gardens water features, and shaded areas.

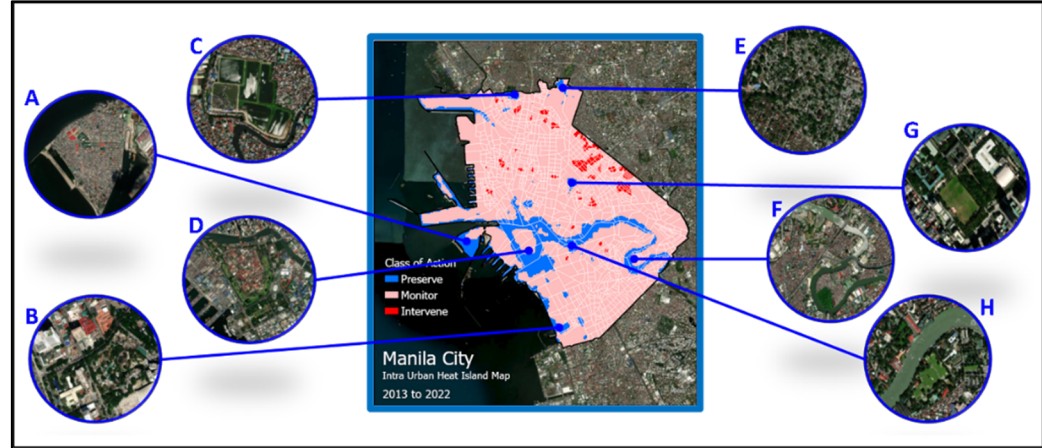

**Figure 15.** Some areas with the "Preserve" Class of Action. (A-H) are the areas highlighted to show their morphologies.

Predominantly, most of the areas shown in Figure 14 exhibit common morphological characteristics. For instance, areas shown in A, F, and H are either surrounded or akin to bodies of water and other water features, while areas shown in C, D, and G contain substantial vegetation and green areas. In addition, areas like B and E, although residential, also contain a decent quantity of trees spread within the area.

In this visual inspection, the two areas have distinguishable features which relate to the surface temperature in the area. Understanding the morphological characteristics of the cold spots (preserve) can help in planning the mitigation strategies needed to improve the thermal condition of the hotspots (intervene).

### 3.6.2. IUHI Class of Action and LULC Indicators Assessment

Overlaying the 2022 maps with the IUHI Map, the average values per class of action are shown in Table 13. It can be observed that the average NDVI values do not provide a clear distinction among the classes of action since the expected cold spots (water bodies and vegetation) have values at the extremes of the index. On the contrary, NDWI and NDBI average values convey the results. For instance, for "preserve", the average NDWI translates to higher water content while the average NDBI shows non-built-up areas. A similar remark can be drawn for "intervene" values where the average NDWI means low water content and the average NDBI falls in the built-up area category.

**Table 13.** Average values of LULC indicators per IUHI class of action.

| Class of Action | Average NDVI | Average NDWI | Average NDBI |
| --- | --- | --- | --- |
| Preserve | 0.209 | 0.089 | −0.090 |
| Monitor | 0.190 | −0.027 | 0.028 |
| Intervene | 0.158 | −0.079 | 0.081 |

Using the same data, we also investigate how the individual index classification is distributed among the IUHI class of action to validate it with the literature. Table 14 provides the distribution of NDVI-based LULC per class of action. It can be observed that areas considered as "preserve" have a higher proportion of water bodies and vegetation while areas considered as "intervene" mostly fall into the urban built-up category.

**Table 14.** Distribution of LULC per IUHI class of action based on NDVI.

| Class of Action | Water Body | Urban Built-Up | Vegetation | Total |
| --- | --- | --- | --- | --- |
| Preserve | 1.76% | 6.11% | 6.24% | 14.10% |
| Monitor | 0.21% | 55.24% | 27.80% | 83.25% |
| Intervene | 0.00% | 2.26% | 0.39% | 2.65% |
| Total | 1.96% | 63.61% | 34.43% | 100.00% |

Table 15 shows the distribution of water content category per IUHI class of action based on NDWI. Based on the proportions, most parts of the areas considered "preserve" have high water content while those for "intervene" have low water content. This shows that the water content of the area has an impact on its surface temperature.

**Table 15.** Distribution of Water Content category per IUHI class of action based on NDWI.

| Class of Action | High Water Content | Low Water Content | Total |
| --- | --- | --- | --- |
| Preserve | 10.07% | 4.03% | 14.10% |
| Monitor | 26.72% | 56.52% | 83.25% |
| Intervene | 0.11% | 2.54% | 2.65% |
| Total | 36.91% | 63.09% | 100.00% |

Table 16 shows the distribution of built-up categories per IUHI class of action based on NDBI. As shown about two-thirds of the "preserve" area occupy non-built-up locations while almost all parts of the "intervene" area are built up. This illustrates the effect of built-up areas such as infrastructures, roads, and buildings that contribute to higher surface temperatures in the city.

**Table 16.** Distribution of Built-up category per IUHI class of action based on NDBI.

| Class of Action | Built-Up | Non-Built-Up | Total |
|---|---|---|---|
| Preserve | 4.19% | 9.91% | 14.10% |
| Monitor | 57.19% | 26.06% | 83.25% |
| Intervene | 2.56% | 0.10% | 2.65% |
| Total | 63.94% | 36.06% | 100.00% |

Based on the observations above, LULC indicators allow us to assess the IUHI maps according to different aspects of the indices. By understanding such categories and how they are related to the IUHI map class of action, the areas can be quantitatively described and later can be used to incorporate mitigation strategies.

### 3.6.3. IUHI Class of Action and High-Resolution Settlement Layer Assessment

The high-resolution settlement layer which consists of population per pixel and settlement categories was also used to assess the IUHI map. The demographic data represent the year 2018 which is the latest available during the conduct of the study.

By superimposing the generated IUHI Class of Action Raster and High-Resolution Settlement Layer containing population per pixel and settlement class, an attribute table is generated. From this attribute table, statistics about the population data and settlement information are taken and summarized in Tables 16 and 17. An example of the attribute table is shown in Figure 16. The object ID represents the corresponding pixel where values related to the attributes are provided. In the Population/Settlement column, population per pixel is shown while those that indicate zero mean a non-settlement pixel.

**Table 17.** Distribution of affected population per IUHI class of action.

| Class of Action | Estimated Affected Population | Population Percentage |
|---|---|---|
| Preserve | 85,601 | 4.92% |
| Monitor | 1,594,166 | 91.55% |
| Intervene | 61,531 | 3.53% |
| Estimated Total Population (2018) | 1,741,298 | 100.00% |

**Figure 16.** Excel Sheet of the superimposed IUHI Class with Population/Settlement Data.

In Table 17, although the percentage of "intervene" areas is small compared to the other IUHI categories, there are still about 61 thousand of the population affected by higher surface temperatures. As Manila is a densely populated city, the population despite its small percentage is still not negligible.

In Table 18, the distribution of settlement categories (from the high-resolution settlement layer data) with IUHI class of action is presented. We can see that about three-fifths (1.70%/2.65%) of the "intervention" area falls on settlement areas. This implies that most of these areas are inhabited by people, which was backed up by the visual inspection in Section 3.4.1. For the "preserve" class of action, most of the areas are non-settlement areas which are mostly vegetated locations, parks, and those near the water features.

**Table 18.** Distribution of settlement category per IUHI class of action.

| Class of Action | Settlement | Non-Settlement | Total |
|---|---|---|---|
| Preserve | 2.37% | 11.73% | 14.10% |
| Monitor | 41.88% | 41.37% | 83.25% |
| Intervene | 1.70% | 0.95% | 2.65% |
| Total | 45.96% | 54.04% | 100.00% |

### 3.6.4. IUHI Class of Action and Land Surface Temperature

To compare the variation of temperature between the cold spots (preserve) and hotspots (intervene), the yearly land surface temperature was calculated for each class of action.

A summary table of the average LST per year per class of action is shown in Table 19. As can be seen, the average difference between the warmest and coldest areas in Manila City is 6.13 °C. The difference through the years has a small deviation wherein the lowest is recorded in 2013 while the highest is in 2017. To better see the trend, a graphical representation of Table 18 is shown in Figure 17.

**Table 19.** Average LST (°C) per year per IUHI class of action.

| | Preserve | Monitor | Intervene | Difference |
|---|---|---|---|---|
| 2013 | 28.56 | 31.87 | 33.94 | 5.38 |
| 2014 | 34.32 | 37.74 | 39.47 | 5.15 |
| 2015 | 37.24 | 41.96 | 44.07 | 6.83 |
| 2016 | 38.88 | 43.19 | 44.78 | 5.90 |
| 2017 | 32.46 | 37.03 | 39.74 | 7.28 |
| 2018 | 33.84 | 37.49 | 40.23 | 6.39 |
| 2019 | 36.00 | 40.81 | 43.12 | 7.12 |
| 2020 | 33.90 | 37.36 | 39.12 | 5.22 |
| 2021 | 36.25 | 40.89 | 42.76 | 6.51 |
| 2022 | 32.91 | 36.79 | 38.39 | 5.48 |
| Average LST | 34.43 | 38.51 | 40.56 | 6.13 |

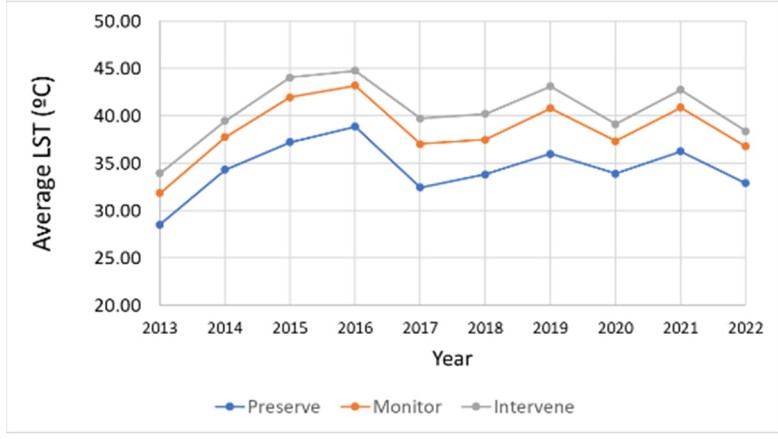

**Figure 17.** Average LST per year per class of action.

3.6.5. Mitigation Strategies for Areas That Need Intervention

With the assessment done in Sections 3.6.1–3.6.4., the differences in temperatures at different urban morphologies were tackled. SDG 11, with its aim to make cities and human settlements inclusive, safe, resilient, and sustainable, can only be realized by not only understanding the city's current situation but also providing means to identify vulnerable areas and implementing solutions to solve existing problems. While the assessment provides information about the presence of intra-urban heat islands in Manila City, this also offers insights into which area in the city policymakers can focus on in offering mitigation strategies. In the analysis, for example, urban settlement and residential areas with narrow streets and sidewalks, asphalted roads and walkways, and concrete commercial spaces can contribute to high surface temperatures, while areas surrounded by and near bodies of water/water features, substantial green spaces/vegetation/trees, and residential areas with decent quantities of trees are places of lower surface temperature. With this in mind, the following mitigation strategies are suggested to help ameliorate the effect of urban heat islands, some of which were adapted from the compendium of strategies by the U.S. Environmental Protection Agency [9].

As part of the local institutional mechanism to address SDG 11, the government can include the following in their priority development initiatives, especially in the identified areas for intervention:

1. Water mist/dry-mist sprayer on pavements and pedestrians. Since the provision of water features may not be possible, mist sprayers can be installed on pavements and pedestrians with the likelihood of people staying or passing by. This inhibits the heat island effect at a low cost and immediately cools the outside air directly [67].

2. Provision of shade structures. Shading can be done in multiple ways, such as with large, canopied trees (which is unlikely based on the assessment) or overhead features to reduce heat buildup in an area. Aside from heat buildup mitigation, it can also be used as protection for people under the heat of the sun.

3. Using cool materials for pavements and roofs. Cool materials are characterized by high solar reflectance and high infrared emittance which result in affecting the temperature of the surface [68]. Replacement of asphalted and concrete roads and pavements with these materials can be done while government-related projects can use cool materials for their roofs and other infrastructures.

4. Provision of cooling centers. Also known as "heat refuge", this includes libraries, community centers, commercial spaces, and other public buildings with cooling systems available to city residents during extreme heat events [69]. Manila City has these spaces already, so additional facilities and designation of such areas is the only requirement.

Additionally, the current densely populated city cannot accommodate extra large-scale trees and vegetation anymore, so the following alternatives can be employed:

5. Conversion of regular walls to green walls. Green walls are partially or completely covered with vegetation and seem lush. They are both beautiful and energizing. Consequently, they absorb warm air, reduce interior and exterior temperatures, and enhance air quality and visual appeal [70]. They are several areas in the city with empty walls but with enough space to convert them to green walls.

6. Plants in plant boxes, road isles, and indoors. One indication of urbanization is the shortage of green spaces [71], so planting in plant boxes, road isles, and indoors can help in improving the thermal landscape without planting trees. Although this cannot provide shading as with a tree canopy, the presence of plants can help in air temperature reduction and evapotranspiration [72]. Manila City still has those spaces for plant boxes and road isles and can encourage its residents to do indoor planting, which is common in the Philippines now.

These are just some of the mitigation strategies applicable to Manila City in its current state. For the attainment of SDG 11 and to address the ill effects of UHI that would result

in a sustainable and livable city, a holistic approach is necessary for implementing such strategies. It should be highlighted that the local government unit including its population plays an important role in this.

## 4. Discussion

The result of this study shows evaluation methods using multiple sources to understand the presence of Intra-Urban Heat Islands in Manila City, Philippines. The satellite data retrieved from Landsat 8 provided distribution maps from 2013 to 2022 which include land surface temperature and LULC indicators such as NDVI, NDWI, and NDBI. More satellite data from MODIS Terra were also obtained to provide point data for land surface temperature data for both day and night. In addition, in-situ data were obtained at Port Area, Manila City, with meteorological data measurements from 2014 to 2018. Finally, raster data containing population density and urban settlement category for 2018 were acquired to represent demographics data for Manila City.

The LST and air temperature data show that beginning in March and continuing through April and May, there is an increasing tendency in the values, whereas values begin to decline in October and continue through January and February, which is similar to the observations in [28,65]. This trend is because March to May is the hot dry season in the Philippines while October to January is rainy and December to February is the cool dry season. In addition, it was found that there is a significant linear relationship between air temperature and land surface temperature based on daily data, while relative humidity shows a weak correlation with the LST data.

In terms of outdoor thermal comfort, a limited analysis was done due to limitations provided by the point measurements of meteorological data in Port Area Manila, City from 2014 to 2018. Despite these limitations, we used the meteorological parameters to estimate the Physiological Equivalent Temperature (PET) thermal index using the RayMan microclimate model. With the calculated PET thermal index values, corresponding physiological stress levels were provided to understand the outdoor thermal comfort. We observed that mild heat stress may be routinely experienced in May, and at certain times in April and June. From July through December, moderate heat stress was seen; however, the thermal comfort zone, where there is no heat stress, did not emerge until January and February. Understanding the thermal comfort in this location may also help us predict the outdoor thermal comfort in other areas of Manila City. It should be noted that the location of Port Area, Manila City is near Manila Bay, which may indicate that the meteorological parameters may not be representative of the whole of Manila City. The calculation of thermal index is calculated based on the meteorological parameters while these meteorological parameters were correlated with land surface temperature. With this, we have associated thermal comfort indirectly with the land surface temperature such that while Port Area, Manila City is not considered as an area for intervention, it still experiences heat stress. Therefore, other areas which are considered areas for intervention are more likely to experience worse thermal stress than Port Area, Manila. This observation and the generated IUHI map can be the basis for selecting additional meteorological stations in areas that may experience worse heat stress, so it can be monitored and provided by mitigation strategies in the future.

Land Use Land Cover (LULC) indicators such as NDVI, NDWI, and NDBI were very useful in understanding the morphological characteristics of Manila City, while their relationship with land surface temperature was also considered. Results of the multivariate analysis show that clusters can be generated based on combinations of these LULC indicators relative to land surface temperature. The clustering findings reveal that values with low NDWI, moderate NDVI, and high NDBI are grouped in the high LST cluster. Low NDWI corresponds to low water content, and high NDBI corresponds to urbanized zones; therefore, this is also predicted. Correlation between LULC indicators and LST shows the link between LST and LULC indicators with their respective slope of linear fit and frequency distribution chart. The data demonstrate a direct association between LST and NDBI at r = 0.361, meaning highly built-up regions have high reported

temperatures. The multivariate analysis supports this finding. LST and NDVI (r = 0.064) and NDWI (r = 0.365) have indirect relationships. A Low Pearson correlation between LST and NDVI implies low temperatures for water bodies and vegetation, whereas mid values imply built-up areas. High water/moisture locations exhibit lower surface temperatures using LST and NDWI. Based on these data, it can be argued that NDWI is a better indication than NDVI for land surface temperature, which agrees with Alexander et al. [66]. NDBI is a good indication for LST, according to the data.

The creation of a space-time cube for LST made spatiotemporal pattern analysis easier. Using the space-time mining tools in ArcGIS Pro, Emerging Hotspot Analysis and Local Outlier Analysis were performed. The resulting reclassified maps of EHSA and LOA were respectively used as input to the suitability analysis model to generate an easy-to-understand Intra-Urban Heat Island (IUHI) class of action map between 2013 to 2022. Such a map contains the class of action (preserve, monitor, and intervene) as well as the administrative boundaries at the city, district, and barangay levels.

In the location assessment, the focus was given to areas to preserve and intervene. Understanding the morphology of "preserve" locations helps in the provision of mitigation strategies for the "intervene" locations. The results show that the highest temperatures are in areas with a concentration of urban settlement areas, buildings, and establishments while those with low temperatures are areas with enough vegetation and near bodies of water. Visual inspection revealed that most "intervene" areas are in the Sampaloc district and university belt. Such an area has a high concentration of universities and colleges while within it are settlement areas, establishments, and concrete roadways which are deemed contributory to the high surface temperature. Knowing this is crucial because aside from its residents, the population in this area swells due to students and employees coming from the nearby province during the daytime. Other intervention areas can be found in the Tondo district, which is home to urban poor communities, while there are also hotspots in the Paco district, which mainly points toward a commercial location. These regions are largely residential, with small streets and sidewalks and a concentration of settlements and dwelling sites. In the regions of concern, initiatives to create an urban soft scape employing trees and plants are limited and scarce. Roads and sidewalks are often constructed with asphalt and concrete, which may contribute to greater surface temperatures. There is also an identifiable commercial area, which seems to have asphalt or concrete companies, buildings, and parking spaces.

On the other hand, "preserve" areas are mostly located in Intramuros, Rizal Park, and sites near the Pasig River banks. Most of the regions have similar physical characteristics. For example, these places are either next to or resembling bodies of water and other water features, while other areas have extensive vegetation and green landscapes. Additionally, residential neighborhoods feature a significant number of trees. Noting these characteristics, mitigation strategies appropriate to the "intervene" areas can be established.

The IUHI class of action was also assessed relative to the corresponding LULC indicator values. While NDVI does not provide a clear distinction among the classes of action, NDVI and NDWI convey their results. For example, the average NDWI for "preserve" indicates a greater water content, but the average NDBI indicates undeveloped lands. Similar observations may be made for "intervene" values when the average NDWI indicates a low water content and the average NDBI falls under the category of "built-up area." Using the same data, we also investigate how the individual index classification is distributed among the IUHI class of action to validate it with the literature. It may be noticed that regions designated as "preserve" have a greater percentage of water bodies and vegetation, higher water content, and occupy non-built-up locations while regions designated as "intervene" are in urban built-up areas with lower water content.

With the high-resolution settlement layer (HRSL), the distribution of the affected population including the settlement category for 2018 was assessed. Upon superimposing the HRSL with the IUHI class of action map, about 61 thousand of the population are affected by higher surface temperatures as indicated in the "intervene" areas. Despite

the small percentage of "intervene" locations compared to the entire Manila City; it is evident that such a small percentage is not negligible due to the city's dense population. In terms of the settlement category, the "intervene" locations are mostly located in settlement areas while the "preserve" locations are in non-settlement areas. Such observation is aligned with what was observed in the visual inspection of locations using high-resolution satellite images.

Summarizing the LST values per year per class of action reveals an average LST for "preserve", "monitor" and "intervene" as 34.43 °C, 38.51 °C, and 40.56 °C, respectively. The result of this study clearly shows differences in temperature within Manila City. With these data, the average difference between cold and warm areas is about 6 °C, just as in the discussion in [20]. As the LST statistics are based on the highest LST readings for each site, it should be understood that the highest LST recorded differentiates 6 °C between specific urban areas. We avoided pixel-based comparison in the overall analysis to evaluate clusters of warm and cold regions appropriate to a city viewpoint and to make the analysis more significant.

Finally, applicable mitigation strategies based on the assessment of cold spots and hotspots in the city were proposed. These strategies support the attainment of SDG 11 in making cities and human settlements inclusive, safe resilient, and sustainable. Such strategies are (1) water mist/dry-mist sprayer in pavements and pedestrians, (2) provision of shade structures, (3) using cool materials for pavements and roofs, (4) provision of cooling center, (5) conversion of regular walls to green walls, and (6) plants in plant boxes, road isles, and indoors.

## 5. Conclusions

This study presents the use of satellite-derived data and meteorological data to assess the presence of an intra-urban heat island in Manila City, Philippines. To address SDG 11 and provide better insights to make cities and human settlements inclusive, safe resilient, and sustainable in terms of UHI, different assessment methods were used and established. The assessment includes (a) understanding the temporal variability of air temperature measurements and outdoor thermal comfort based on meteorological data, (b) comparative and correlative analysis between common LULC indicators (NDVI, NDBI, and NDWI) to LST, (c) spatial and temporal analysis of LST using spatial statistics techniques, and (d) generation of an intra-urban heat island (IUHI) map with a recommended class of action using a suitability analysis model. Finally, the areas that need intervention are compared to the affected population, and suggestions to enhance the thermal characteristics of the city and mitigate the effects of UHI were established. Results show that there exists a clear difference between cold and warm areas within Manila City. Overall, residential areas, asphalted and concrete roads and walkways, and some commercial establishments and buildings exhibit higher surface temperatures compared to areas with vegetation and near bodies of water. Based on the results, mitigation strategies applicable to Manila City were proposed to improve the areas which need intervention.

In the future, we plan to realize these strategies by partnering with the local government unit to implement these proposed measures. We also advise providing additional meteorological stations to some of the hotspots, to understand outdoor thermal comfort in Manila City better. In addition, the methods used in this study can also be used in other cities as well as municipalities that require assessment due to the presence of intra-urban heat islands.

**Author Contributions:** Conceptualization, M.A.P., M.C. and T.Y.; methodology, M.A.P., M.C. and T.Y.; software, M.A.P.; validation, M.A.P., M.C. and T.Y.; formal analysis, M.A.P., M.C. and T.Y.; investigation, M.A.P.; resources, M.A.P.; data curation, M.A.P.; writing—original draft preparation, M.A.P.; writing—review and editing, M.C. and T.Y.; visualization, M.A.P.; supervision, M.C. and T.Y.; project administration, M.A.P.; funding acquisition, M.A.P. and M.C. All authors have read and agreed to the published version of the manuscript.

**Funding:** This work was funded and supported by Adamson University, Kyushu Institute of Technology, and the Department of Science and Technology—Science Education Institution (DOST-SEI) through the STAMINA4Space Program of the University of the Philippines-Diliman.

**Data Availability Statement:** Not applicable.

**Acknowledgments:** The authors would like to thank Adamson University, Kyushu Institute of Technology, and the Department of Science and Technology—Science Education Institution (DOST-SEI) through the STAMINA4Space Program of the University of the Philippines-Diliman for supporting and funding this research. Moreover, we thank the following who one way or another helped in the realization of this work: Climate and Agrometeorological Data Section (CADS), Climatology and Agrometeorology Division (CAD), Philippine Atmospheric, Geophysical and Astronomical Services Administration (PAGASA) for the weather data; Maria Fe B. Abalos, Knowledge Management and Communications Division, Information Technology and Dissemination Service, Philippine Statistics authority for Philippine statistical data including the population for Manila City; Julius M. Judan, SSED-Ground Receiving Station, DOST-Advanced Science and Technology Institute for giving access to remote sensing data and satellite images; Joven Javier, for assisting in communicating with the government agencies such as PAG-ASA and DOST-ASTI; Joseph Ronquillo and Anna May Ramos for the statistical analysis, Ronnie Serfa Juan and Evelyn Raguindin for revision comments; and BIRDS-4 satellite project members for the support.

**Conflicts of Interest:** The authors declare no conflict of interest.

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
