# Peer review of "Assessment of Intra-Urban Heat Island in a Densely Populated City Using Remote Sensing: A Case Study for Manila City"

_remotesensing, doi:10.3390/rs14215573_

Round 1
Reviewer 1 Report
The paper entitled "Assessment of Intra-Urban Heat Island in a Densely
Populated City using Remote Sensing: A Case Study for Manila City" presents the evaluation of the UHI in Manila and its relationship with the urban landscape changes to generate a database for urban management plans. The methodology section is significant and detailed, while the results are presented in a high-quality way. The aim and methods are in line with the scope of the journal. The paper contributes to a steamily well-developed investigation in urban climate studies.
Therefore, I recommend that some minor revisions be considered before publication in the journal.
Therefore, I recommend that some minor revisions be considered before publication in the journal.
The introduction section could be reduced in length. It would be advisable that the authors highlight the objective with clarity.
Figure 1 needs to be improved. The coordinate system is not adequate. It lacks latitude and longitude references, as well as legend. The scale bar in the second image is crucial to be included. It does not have a spatial reference, so it isn't easy to understand how many pixels or the size of the satellite images are needed for the study.
In the methodology section could be relevant that the authors include more details about the methods and techniques used to analyze and process/gap-filled the missing information related to in situ measurements.
Why do the authors use MODIS information? Are these products adequate to study the intra-urban heat island? It is essential to notice that urban studies are relevant in a micro-scale frame with high-resolution data.
Why do authors correlate MODIS information with in-situ data? Are these variables coherently correlated? Is it possible to connect these variables? Is it climatically possible?
There are some parts in the methodology section that need to be improved. It is necessary to include more details in some methods (i.e., The spatial and spectral resolution, the number of pixels that covers the City, and the reasons for correlating variables, among others). Moreover, the authors must reduce some of the information presented in this section. Some figures and tables should be avoided.
Figure 5 needs to be presented at the beginning of the Methodology section to introduce every step that the authors consider for their research. Moreover, its relocation could be helpful for generating a clear document.
In the results, Figures 9 and 10 need to include coordinate systems. The same pattern is found in the rest of the images. In my opinion, the word "legend" is not necessary.
It is remarkable the excellent quality of the discussion and conclusion sections.
Author Response
Please see attachment for the response.
Updated Manuscript Link: https://drive.google.com/file/d/19uHfnhj0l1h4ypl3IkwTmLMxp07z2-iY/view?usp=sharing

Reviewer 2 Report
This manuscript evaluated intra-urban heat islands(IUHI) in Manila city, Philippines using satellite-derived and meteorological data. The originality of this manuscript lies on the use of space-time pattern mining, and population and settlement data to assess the IUHI using remote sensing data. However, the following things need to be revised.
First, at the bottom line in Table 8 on page 13, "~a cluster of high~" should be changed to "~a cluster of low~".
Second, in line 548 on page 15, Table 13 should be change to Table 10.
Third, you should justify how satellite-derived surface temperatures can represent the IUHI in Manila city. That's because Landsat 8 passes over the Manila city at 10:00 am +/- 15 minutes (mean local time). The IUHI can be measured well afternoon rather than morning in Manila city.
Fourth, you should explain how you quantified the affected population estimates and settlement in Tables 16 and 17.
Fifth, it is not clear why you used both emerging hotspot analysis and local outlier analysis to generate the IUHI map. If I use either one, is there any difference in space-time pattern for the IUHI map?
Author Response

(The authors gave the same response as above.)
